# Protection of COVID-19 vaccination and previous infection against Omicron BA.1, BA.2 and Delta SARS-CoV-2 infections

Stijn P. Andeweg [1], Brechje de Gier[1], Dirk Eggink[1], Caroline van den Ende[1], Noortje van Maarseveen[2,3], Lubna Ali[2], Boris Vlaemynck[4], Raf Schepers[4], Susan J. M. Hahné[1], Chantal B. E. M. Reusken [1], Hester E. de Melker[1], Susan van den Hof [1] & Mirjam J. Knol [1]✉

Given the emergence of the SARS-CoV-2 Omicron BA.1 and BA.2 variants and the roll-out of booster COVID-19 vaccination, evidence is needed on protection conferred by primary vaccination, booster vaccination and previous SARS-CoV-2 infection by variant. We employed a test-negative design on S-gene target failure data from community PCR testing in the Netherlands from 22 November 2021 to 31 March 2022 (n = 671,763). Previous infection, primary vaccination or both protected well against Delta infection. Protection against Omicron BA.1 infection was much lower compared to Delta. Protection was similar against Omicron BA.1 compared to BA.2 infection after previous infection, primary and booster vaccination. Higher protection was observed against all variants in individuals with both vaccination and previous infection compared with either one. Protection against all variants decreased over time since last vaccination or infection. We found that primary vaccination with current COVID-19 vaccines and previous SARS-CoV-2 infections offered low protection against Omicron BA.1 and BA.2 infection. Booster vaccination considerably increased protection against Omicron infection, but decreased rapidly after vaccination.

[1] Center for Infectious Disease Control, National Institute for Public Health and the Environment (RIVM), Bilthoven, the Netherlands. [2] Saltro Diagnostic Center for Primary Care, Utrecht, The Netherlands. [3] Department of Medical Microbiology, University Medical Center Utrecht, Utrecht, the Netherlands. [4] SYNLAB, Heppignies, Belgium. ✉email: mirjam.knol@rivm.nl

Since November 2021 the Omicron BA.1 SARS-CoV-2 variant (Nextstrain clade 21K, Pango lineage B.1.1.529) has rapidly spread causing high numbers of infections across Europe, due to higher transmissibility and immune escape[1]. In the Netherlands, the Omicron variant was first detected end of November 2021[2]. Several studies indicate that Omicron variants cause less severe disease than the Delta variant (Nextstrain clade 21A, Pango lineage B.1.617.2), also when adjusted for infection- and vaccine-induced immunity[3–5]. However, this characteristic could be outweighed by large numbers of infections because of greater transmission resulting in increased pressure on hospital care. First, the Omicron BA.1 variant emerged; from January 2022 a new variant from Omicron, BA.2 (Nextstrain clade 21L), started to replace Omicron BA.1.

In vitro assays have shown largely reduced neutralisation of both Omicron variants by pre-Omicron convalescent sera and by sera of vaccinated individuals[6,7]. Only small differences in neutralisation are observed between the Omicron BA.1 and BA.2 variants[7,8]. Early studies on vaccine effectiveness showed very limited protection of primary COVID-19 vaccination against Omicron infection[9]. Booster vaccination increases protection, although to a smaller extent than against Delta. Also, protection against symptomatic COVID-19 seems to wane rapidly[10]. Data from England showed no evidence for reduced vaccine effectiveness against symptomatic infection with BA.2 as compared to BA.1[11]. The rate of reinfections with Omicron is larger than seen with other variants, indicating that previous infections with another variant do not provide sufficient protection against Omicron[12,13]. The level of protection against infection induced by the combination of vaccination and previous infection is largely unknown. Vaccine effectiveness against severe disease seems to be better preserved and restored to around 90% shortly after booster vaccination[10].

The Dutch COVID-19 vaccination program started in January 2021, first targeting residents of long-term care facilities and health care workers, and then following an age-dependent rollout starting with the eldest cohorts. At the end of summer 2021, all persons of 12 years and older had had the opportunity to complete primary vaccination. At the end of January 2022 individuals aged 5–11 years became eligible for vaccination. The coverage of full primary vaccination was 86% on April 3 2022 in adults 18 years or older; for children aged 12–17 years this was 70%; for children aged 5–11 years this was 6%[14]. In November 2021, the booster vaccination campaign targeting adults started in the Netherlands. On January 9 2022, 45% of the adult population had received a booster vaccination and on April 3 2022 this was 63%[15]. Comirnaty (BNT162b2, BioNTech/Pfizer, 77% of primary vaccination doses), Spikevax (mRNA-1273, Moderna, 8%), Vaxzevria (ChAdOx1, AstraZeneca, 11%) and Janssen vaccine (4%) have been used for primary vaccination in the Netherlands[16]. Comirnaty (49% of all booster doses) and Spikevax (51%) have been used for booster vaccination.

We previously showed with a case-only design that infection- and vaccine-induced protection of the primary series was much reduced for Omicron BA.1 infection compared with Delta[13]. Here we use a test-negative design to estimate the effects of primary and booster vaccination and the previous infection on the protection against infection with Omicron BA.1 compared to Delta and with Omicron BA.2 compared to BA.1, during two periods when both variant combinations circulated in the Netherlands. S-gene target failure (SGTF) was used as a proxy for Omicron BA.1 infection and non-SGTF was used as a proxy for Delta or Omicron BA.2 infection, dependent on time period. We assess the protection conferred by primary and booster vaccination, with and without previous infections, against infection with the Delta, Omicron BA.1 and BA.2 SARS-CoV-2 variants by time since vaccination or previous infection and by age group. To this end, we employed data from 671,763 community-based SARS-CoV-2 tests performed between November 22, 2021, and March 31, 2022.

## Results

**Study population.** Between 22 November 2021 and 31 March 2022, 2,264,333 samples were tested at one of the two diagnostic laboratories that performed SGTF-PCR (Fig. 1). After exclusions, 354,653 tests were included in the analysis for the Delta-Omicron BA.1 comparison and 317,110 for Omicron BA.1-BA.2 (Figure S1). In the Delta-Omicron BA.1 cohort, 300,849 (84.8%) were negative, 39,889 (11.2%) were positive for Delta and 13,915 (3.9%) were positive for Omicron BA.1 (Table 1). Among positive tests, the proportion of SGTF increased from 0.2% on 28 November 2021 to 91.0% on 7 January 2022, with an earlier increase in the 18–29 age group (Figure S2). Whole-genome sequencing (WGS) of 485 cases indicates a positive predictive value (PPV) of 1.00 (132 out of 132) for SGTF and Omicron BA.1 and 1.00 (353 out of 353) for non-SGTF and Delta in this cohort (Table S1). In the Omicron BA.1-BA.2 cohort, 207,553 (65.5%) tested negative, 67,887 (21.4%) were positive for Omicron BA.1 and 41,670 (13.1%) for BA.2 (Table 1). The proportion of SGTF among cases decreased in this time period from 89.6% on 26 January 2022 to 4.8% on 31 March 2022 (Figure S2). In this cohort sequencing of 288 cases demonstrated a PPV of 1.00 (158 out of 158) for SGTF and Omicron BA.1 and 0.98 (128 out of 130) for non-SGTF and Omicron BA.2 (Table S1).

**Delta-Omicron BA.1 cohort.** Previous infection, primary vaccination, or both protected well against Delta infection, with a relative reduction of 76% (95% confidence interval (CI): 73–79), 71% (95% CI: 69–73) and 92% (95% CI: 87–95, infection first estimate), respectively, at 7+ months after vaccination or previous infection (Fig. 2A, Data S1). Booster vaccination increased the overall protection against Delta infection to 94% (95% CI: 94–96), or 99% (95% CI: 95–100) in persons with also a previous infection. Protection by previous infection or primary vaccination was much lower against Omicron BA.1 infection, with overall relative reductions of 13% (95% CI: 4–21) and 22% (95% CI: 18–26) after previous infection or primary vaccination, respectively. Persons who received primary vaccination and had a previous infection were better protected than after only primary vaccination or only previous infection, with an overall relative reduction of 49% (95% CI: 41–55, first infection, then primary vaccination) and 52% (95% CI: 45–59, first start primary vaccination, then infection). Recent booster vaccination increased protection against Omicron BA.1 infection to 58% (95% CI: 55–62), or 68% (95% CI: 58–75) in persons without and with the previous infection, respectively, but it was remarkably lower than against Delta infection (Fig. 2A, Data S1). The overall relative protection afforded by booster vaccination compared with primary vaccination was 75% (95% CI: 72–79) against Delta infection and 46% (95% CI: 42–50) against Omicron BA.1 infection (Table S2).

**Omicron BA.1-BA.2 cohort.** Across all vaccination and previous infection statuses protection against Omicron BA.1 and BA.2 infection was comparable (Fig. 2B, Data S1). For Omicron BA.1 and BA.2 infection, the relative reduction of infection compared to naive at 7+ months after primary vaccination was 39% (95% CI: 36–42) and 32% (95% CI: 29–36) and at 1 month after booster vaccination 69% (95% CI: 67–70) and 61% (58–63), respectively. The previous infection 7+ months ago offered protection of 34% (95% CI: 31–38) against BA.1 infection and 38% (95% CI: 34–43) against BA.2 infection. Previous infections in the Omicron BA.1-BA.2 cohort with a short interval can include Omicron BA.1 previous infections (Fig. 1). Again, the combination of vaccination and

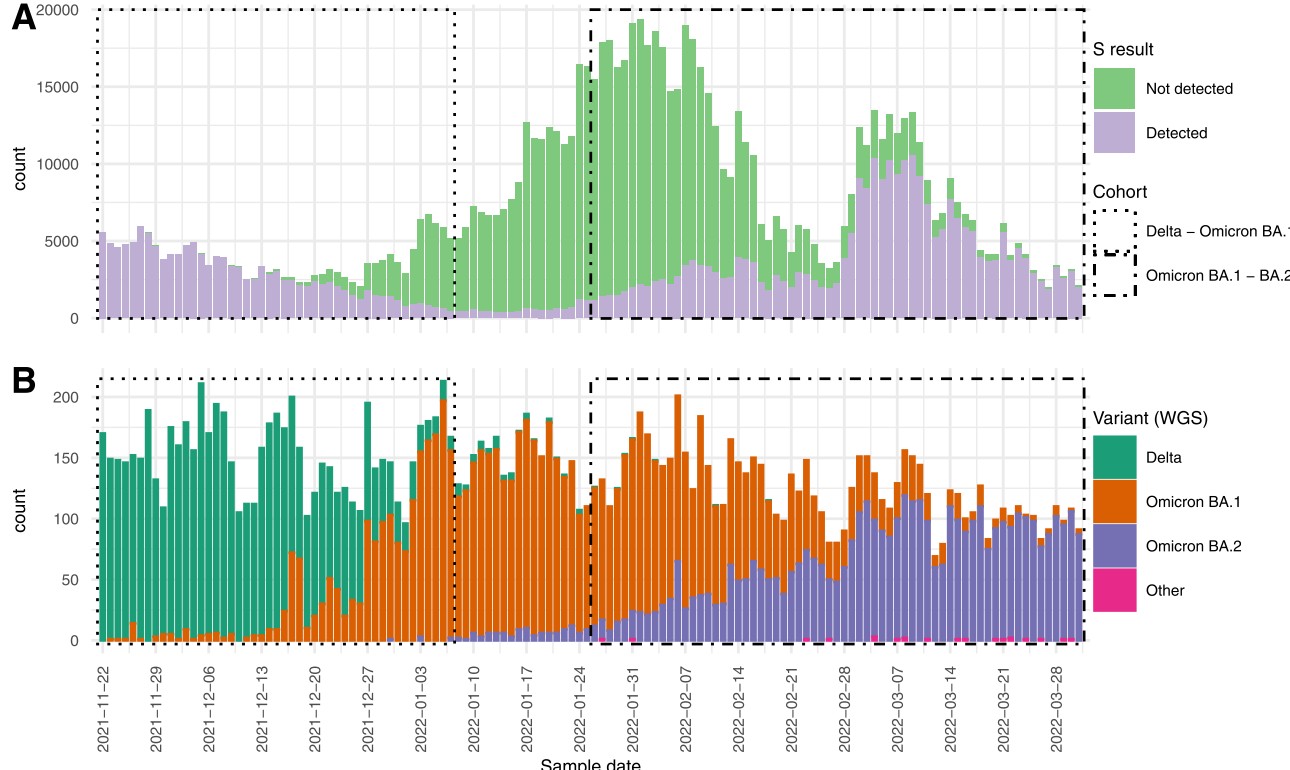

**Fig. 1 S result and WGS typed variant found in community surveillance. A** Number of S-gene target failure (SGTF) and non-SGTF positive tests over time ($n = 986{,}974$). **B** Whole-genome sequencing (WGS) typed variant found in national genome surveillance of community testing samples in the Netherlands ($n = 17{,}907$). Boxes indicate the Delta-Omicron BA.1 cohort (dotted line) and Omicron BA.1-BA.2 cohort (dot-dashed line).

infection gave higher protection than either one alone, with an overall reduction of 69% (95% CI: 66–72, first infection, then primary vaccination) and 85% (95% CI: 84–87, first start primary vaccination, then infection) against Omicron BA.1 and 72% (95% CI: 68–75, first infection, then primary vaccination) and 85% (95% CI: 84–87, first start primary vaccination, then infection) against Omicron BA.2. The overall relative protection afforded by booster vaccination against Omicron BA.1 and BA.2 infection compared with primary vaccination was 48% (95% CI:47–49) and 40% (95% CI: 38–43), respectively (Table S2).

**Waning protection.** Protection against Delta, Omicron BA.1 and BA.2 infection waned with time since primary vaccination or infection (Fig. 2). For Omicron BA.1 and BA.2 infection, the relative reduction decreased from over 70% shortly after primary vaccination or infection to 32–39% at 30 weeks or more after primary vaccination or infection. For Delta infection, the relative reduction decreased from 78–95% to 71–84% over time since primary vaccination or infection. Also in persons with both primary vaccination and previous infection, the waning of protection was observed by time since vaccination or infection. In the first four months after booster vaccination a decrease in effectiveness against Omicron BA.1 and BA.2 was observed, from 61–69% in the first month after vaccination to 47–51% in the fourth month. This large decrease was not observed in individuals with both booster vaccination and previous infection (Fig. 2B).

**Age.** In general, similar patterns were seen across age groups, with higher protection from previous infection and vaccination against Delta infection than Omicron BA.1 infection, and similar protection against Omicron BA.1 and BA.2 infection (Fig. 3, Data S2). However, we observed some differences between age

groups. The protection conferred by the previous infection against Omicron BA.1 and BA.2 was significantly higher for children 0–11 and 12–17 years compared to persons aged 18–59 (Fig. 3, Data S3). The effectiveness of primary vaccination against Delta infection decreased with increasing age. In contrast, protection from vaccination (alone or in combination with the previous infection) against Omicron BA.1 and BA.2 infection was for some of the comparisons significantly higher in those aged 60 years and older compared to 18–29 year olds (Fig. 3, Data S3).

## Discussion

We assessed the protection afforded by the previous infection, and primary and booster vaccination with and without previous infection against Delta, and Omicron BA.1 and BA.2. Our analyses show a substantial reduction in the protection conferred by previous SARS-CoV-2 infections and/or vaccination against the Omicron BA.1 variant, as compared to the Delta variant. The protection conferred by vaccination and previous infection was comparable between Omicron BA.1 and BA.2. Booster vaccination increased effectiveness against Omicron, but to a lesser extent than against Delta, resulting in a much lower booster vaccine effectiveness for Omicron compared with Delta. Moreover, three months after booster vaccination protection against BA.1 and BA.2 had decreased sharply.

The modest vaccine effectiveness against Omicron infection we found is comparable to estimates from other studies[10,17–19]. Similar to our findings, others have also shown considerable waning of the effectiveness of primary vaccination against both Delta and Omicron infection[10,17,19]. Data from the UK show an initial increase of effectiveness after booster vaccination to 50% against Omicron infection and a rapid decrease over time to 30% at 4–6 months after booster vaccination[10]. Other studies found, in line with our findings, no apparent differences in the effect of vaccination against BA.1 versus BA.2 infection[11,20].

**Table 1 Characteristics of persons testing negative for SARS-CoV-2, testing positive for Delta SARS-CoV-2 (S-gene target detected, cohort Delta-Omicron BA.1), Omicron BA.1 (S-gene target not detected), and BA.2 SARS-CoV-2 (S-gene target detected, cohort Omicron BA.1-BA.2), the Netherlands, 22 November 2021 – 7 January 2022 (Cohort Delta-Omicron BA.1, n = 354,653) and 26 January – 31 March 2022 (Cohort Omicron BA.1-BA.2, n = 317,110).**

| | | Cohort Delta-Omicron BA.1 | | | Cohort Omicron BA.1-BA.2 | | |
|---|---|---|---|---|---|---|---|
| | | Negative, n (%) | Delta, n (%) | Omicron BA.1, n (%) | Negative, n (%) | Omicron BA.1, n (%) | Omicron BA.2, n (%) |
| Total | | 300,849 | 39,889 | 13,915 | 207,553 | 67,887 | 41,670 |
| Age (years) | <=11 | 58,114 (19.3) | 9162 (23.0) | 891 (6.4) | 23,728 (11.4) | 7550 (11.1) | 3211 (7.7) |
| | 12-17 | 24,497 (8.1) | 2290 (5.7) | 792 (5.7) | 13,555 (6.5) | 6567 (9.7) | 1846 (4.4) |
| | 18-29 | 54,280 (18.0) | 5399 (13.5) | 5242 (37.7) | 42,840 (20.6) | 14,518 (21.4) | 9145 (21.9) |
| | 30-59 | 125,575 (41.7) | 16,862 (42.3) | 5843 (42.0) | 100,797 (48.6) | 33,469 (49.3) | 20,464 (49.1) |
| | 60+ | 38,383 (12.8) | 6176 (15.5) | 1147 (8.2) | 26,633 (12.8) | 5783 (8.5) | 7004 (16.8) |
| Sex | Male | 140,874 (46.8) | 19,879 (49.8) | 6614 (47.5) | 92,192 (44.4) | 31,202 (46.0) | 19,009 (45.6) |
| | Female | 159,975 (53.2) | 20,010 (50.2) | 7301 (52.5) | 115,361 (55.6) | 36,685 (54.0) | 22,661 (54.4) |
| Vaccination and previous infection status | Naive | 90,945 (30.2) | 21,042 (52.8) | 3440 (24.7) | 42,182 (20.3) | 24,205 (35.7) | 10,437 (25.0) |
| | Previous infection, unvaccinated | 12,691 (4.2) | 638 (1.6) | 739 (5.3) | 17,317 (8.3) | 5498 (8.1) | 2898 (7.0) |
| | Primary vaccination | 173,095 (57.5) | 17,777 (44.6) | 8134 (58.5) | 32,577 (15.7) | 14,930 (22.0) | 6871 (16.5) |
| | Booster | 12,792 (4.3) | 215 (0.5) | 1004 (7.2) | 95,872 (46.2) | 20,777 (30.6) | 19,499 (46.8) |
| | First start primary vaccination, then infection | 3406 (1.1) | 76 (0.2) | 240 (1.7) | 7020 (3.4) | 691 (1.0) | 539 (1.3) |
| | First infection, then primary vaccination | 7002 (2.3) | 139 (0.3) | 293 (2.1) | 2688 (1.3) | 597 (0.9) | 315 (0.8) |
| | Previous infection, booster | 918 (0.3) | 2 (0.0) | 65 (0.5) | 9897 (4.8) | 1189 (1.8) | 1111 (2.7) |
| Vaccine | Comirnaty | 147,480 (49.0) | 13,504 (33.9) | 6986 (50.2) | 98,728 (47.6) | 27,008 (39.8) | 18,250 (43.8) |
| | Spikevax | 21,899 (7.3) | 1262 (3.2) | 1189 (8.5) | 38,660 (18.6) | 8160 (12.0) | 7778 (18.7) |
| | Vaxzevria | 14,535 (4.8) | 2018 (5.1) | 436 (3.1) | 2422 (1.2) | 644 (0.9) | 491 (1.2) |
| | Janssen | 12,807 (4.3) | 1412 (3.5) | 1087 (7.8) | 3915 (1.9) | 1508 (2.2) | 774 (1.9) |
| | None | 103,636 (34.4) | 21,680 (54.4) | 4179 (30.0) | 59,499 (28.7) | 29,703 (43.8) | 13,335 (32.0) |
| | Unknown | 492 (0.2) | 13 (0.0) | 38 (0.3) | 4329 (2.1) | 864 (1.3) | 1042 (2.5) |
| Median interval (IQR) vaccination or previous infection sample date and current sample date (days) | Primary vaccination | 148 (124-175) | 156 (132-184) | 159 (143-184) | 189 (162-217) | 183 (159-206) | 205 (173-231) |
| | Booster | 15 (12-22) | 14 (10-19) | 15 (11-23) | 45 (31-63) | 38 (28-53) | 64 (48-78) |
| | Previous infection | 258 (184-352) | 328 (238-387) | 283 (177-388) | 236 (104-384) | 293 (148-412) | 258 (118-411) |
| Symptoms at time of test request | Symptoms reported | 199,026 (66.2) | 27,906 (70.0) | 10,429 (74.9) | 155,486 (74.9) | 56,976 (83.9) | 37,657 (90.4) |
| | No symptoms reported | 100,150 (33.3) | 11,719 (29.4) | 3418 (24.6) | 51,144 (24.6) | 10,619 (15.6) | 3873 (9.3) |
| | Unknown | 1673 (0.6) | 264 (0.7) | 68 (0.5) | 923 (0.4) | 292 (0.4) | 140 (0.3) |

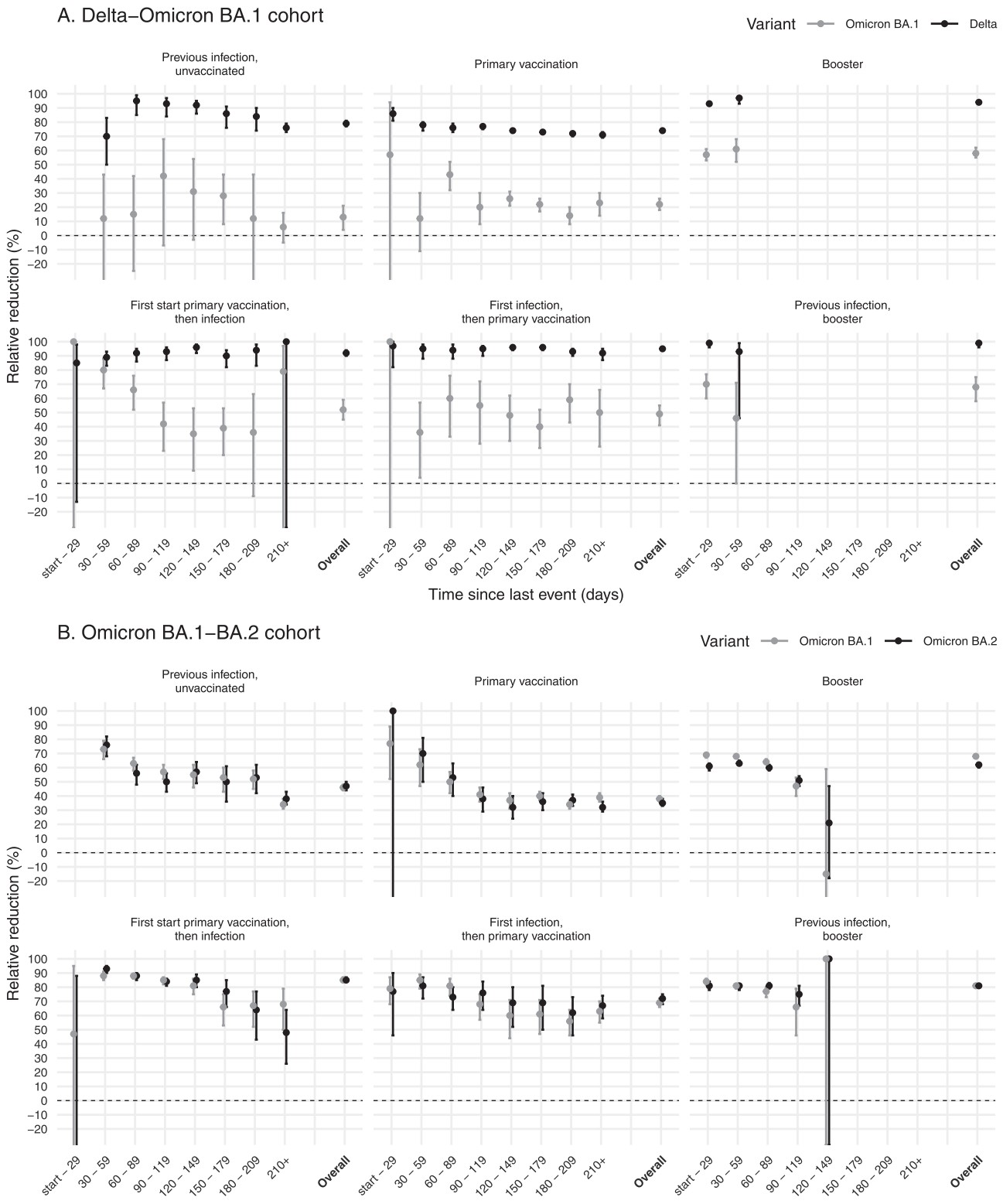

**Fig. 2 Relative reduction in infections for different vaccination and previous infection statuses.** Relative reduction in infections after previous infection, primary vaccination, booster vaccination, or combinations of previous infection and vaccination, compared with naïve status ((1-odds ratio (OR)) * 100), by time since last event and overall in persons aged 18 and older, for cohort Delta-Omicron BA.1 (**A**, n = 258,907) and cohort Omicron BA.1-BA.2 (**B**, n = 260,653). Error bars represent 95% confidence intervals.

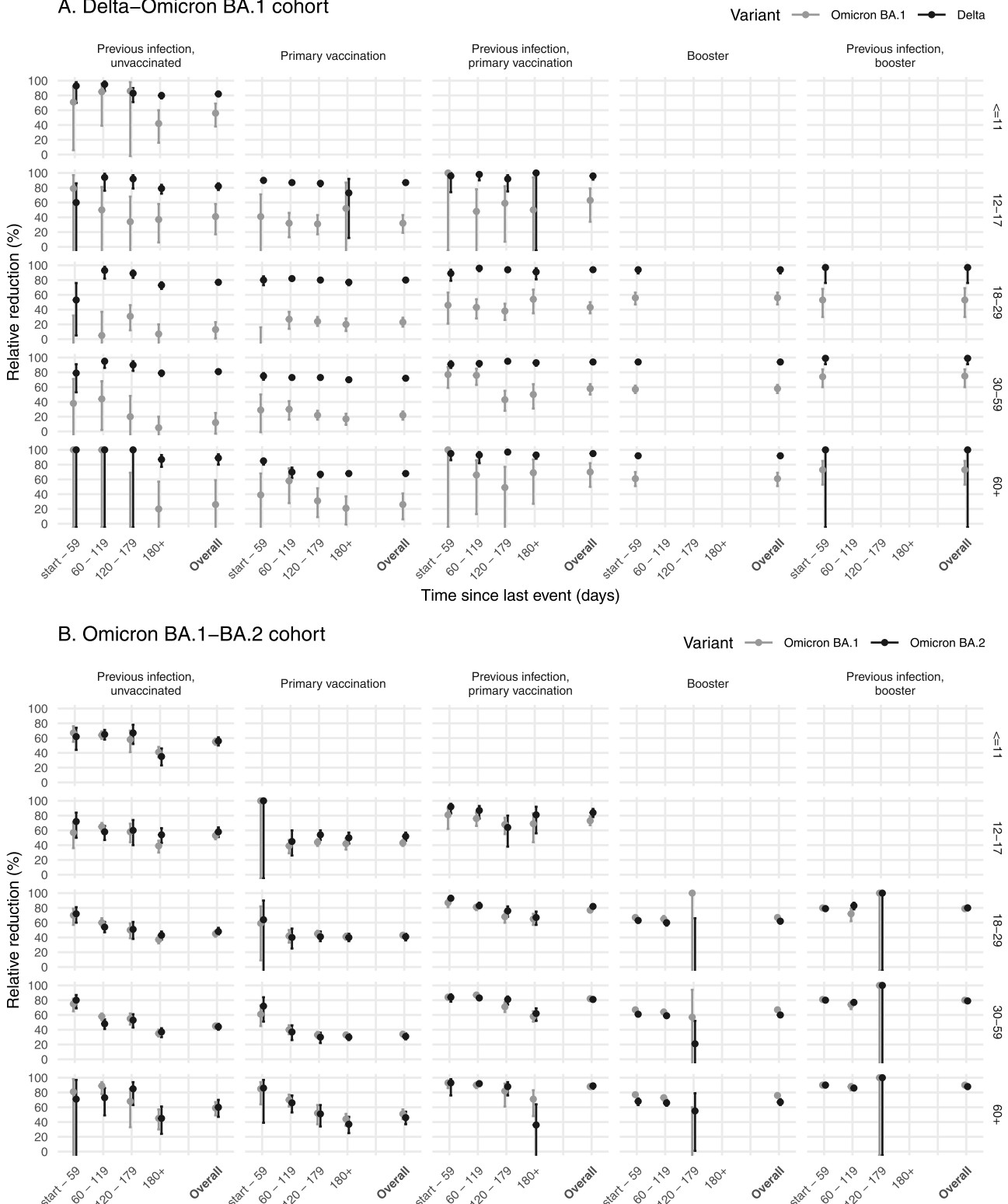

**Fig. 3 Relative reduction in infections for different vaccination and previous infection statuses by age.** Relative reduction in infections after previous infection, primary vaccination, booster vaccination, or combination of previous infection and vaccination, compared with naïve status ((1-odds ratio (OR)) * 100), by time since last event and overall, and by age group. The columns indicate the vaccination and previous infection status and the rows display the age group in years, for cohort Delta-Omicron BA.1 (**A**, $n = 354,596$) and cohort Omicron BA.1-BA.2 (**B**, $n = 316,973$). Error bars represent 95% confidence intervals.

Our results showed loss of protection against Omicron infection conferred by the previous infection with a different variant, which is in line with an observed increase in re-infection cases since the emergence of Omicron[12,21,22]. In the Netherlands, the percentage of new positive tests that were re-infections rose from around 3% during the Delta-dominated period to 12–13% in the first months of 2022[23]. A preprint analyzing reinfections in Qatar found that protection of the previous infection against Omicron infection was 62%[24], which was considerably lower than the protection afforded by the previous infection in preventing infection with Delta, but higher than the protective effect in our study, which was 45–50%. Another preprint (from the same authors) reported that protection from the previous infection was similar for BA.1 and BA.2, in line with our findings[25].

In our study, the combination of both previous infection and primary vaccination provided more protection against Omicron than either of those alone. The highest levels of protection were observed in recently boosted individuals with a previous infection. Protection estimates for persons with first infection then vaccination and first vaccination then infection were quite comparable, suggesting that the sequence of vaccination and infection did not influence the level of protection. A recent study found broadly neutralizing antibodies against several variants of concern, not including Omicron, in sera from vaccinated plus infected individuals regardless of the sequence of vaccination and infection[26]. Another recent study found that the number of immunizing events (vaccinations and/or infections) correlates with the quality and breadth of the neutralizing antibody response, including against Omicron[27]. Our results are in line with these studies and indicate that reinfections and breakthrough infection during the Omicron wave could contribute to broader immunity in the population, also against future variants. Of note, in our study the previous infections in persons with vaccination after infection will likely mostly have been infections by wildtype SARS-CoV-2 or the Alpha variant (although especially in the younger population start of vaccination coincided with a Delta peak in Summer 2021), while previous infections in persons with infection after vaccination will likely have been infections by the Delta variant. This means that not only the sequence of vaccination and infection is different between these groups but most likely also the variant of infection, which makes the groups not directly comparable and the results more difficult to interpret.

We observed differences between the Delta-Omicron BA.1 and Omicron BA.1-BA.2 cohorts. Generally, the BA.1 protection estimates are higher in the second cohort. We do not have a conclusive explanation for the observed variation. However, this could be due to differences between the cohorts related to the calendar period. First, the testing advice changed on 18 January 2022 (in between the cohorts), where recently infected and boostered individuals are no longer advised to seek asymptomatic testing after high-risk contact. Second, previous infections have been accrued in different time periods leading to different distributions of variants causing the previous infection. Third, in the Omicron BA.1-BA.2 cohort the SARS-CoV-2 incidence was higher as reflected in the increased positivity (Table 1). The resulting larger numbers of BA.1 positive tests included in the latter cohort allowed for more precise, and possibly more valid, estimates.

Generally, we found similar trends across age groups with higher protection from vaccination and previous infection against Delta infection than Omicron infection and similar protection against Omicron BA.1 versus BA.2 infections. Noticeably, we found better protection from previous infection in age groups <18 years compared with persons aged 19–59. As of yet, we do not have an explanation for this finding, which merits further

research. Unexpectedly, we found some indication of higher protection from vaccination (with the previous infection) against Omicron infection in persons aged 60 years or older compared to in younger adults. There is no clear biological explanation for this observation. Other differences between age groups could play a role, for example, differences in misclassification of previous infections, in exposure, in vaccine brands used and in the timing of vaccination.

Our study has some limitations. We use observational data from community testing registries to estimate protection from vaccination and previous infection and thereby we implicitly assume that unvaccinated and vaccinated persons, and persons with a previous infection are similar with respect to, for example, exposure to SARS-CoV-2, which may not be the case. Still, we think comparisons between variants within the two cohorts, and herein comparisons over time since vaccination or infection are valid. In addition, our results are comparable with results from other countries. Also testing behavior may be different between unvaccinated and vaccinated persons, and persons with a previous infection. A test-negative study has the advantage of only including individuals coming forward for testing, limiting bias from differential testing behavior. Testing behavior could also be different for Omicron vs Delta infection because of different symptomatology. A sensitivity analysis including only symptomatic persons gave similar results (Data S1), which is reassuring, although we were not able to look at severity or duration of symptoms. Vaccination status is self-reported and may have led to some misclassification, although we do not expect a differential misclassification effect by variant. Also, individuals with a previous infection before onset of primary vaccination were given the choice to only receive one dose of a two-dose schedule to complete the primary series. This group could have been misclassified as having received a primary series while the second dose was a booster dose. In around 6% of the Dutch population having received two doses the second dose was a booster after a one-dose primary series. This misclassification may have resulted in overestimation of the VE shortly after completion of a primary series. Another source of misclassification is the use of community testing data to determine previous infection status. A relevant share of previous infections will be missing from this dataset, e.g. because of restrictive testing policy in the early months of the pandemic. For children below the age of 12, testing was not strongly encouraged for a long time, and the mild and larger share of asymptomatic disease course in this age group will further contribute to a large share of undetected infections. This misclassification will have led to an underestimation of the effect of a previous infection. Further, (non-)SGTF is not a perfect variant indicator and the predictive value is affected by changes in incidence of the different variants. Therefore, only tests were included from periods in which the PPV of SGTF (compared with WGS) to discern variants was >85% to limit misclassification. Therefore we do not expect this to meaningfully bias results.

In this study, we used data from community testing, mostly covering persons with mild infections at the time of testing. Vaccine effectiveness against severe COVID-19 is higher than against infection, also for the Omicron variant. A UK study found vaccine effectiveness against hospitalization of 56% at 175 or more days after the second dose, and 91% shortly after the booster dose, decreasing to 80% after 70–104 days after the booster dose[10]. These estimates are lower than against severe COVID-19 by the Delta variant. However, the risk of a severe disease course after Omicron infection has been shown to be lower than after Delta infection[3–5]. This has led to lower burden on critical care than in previous waves despite immune escape by the Omicron variants.

In conclusion, we showed that primary vaccination with the current COVID-19 vaccines and pre-Omicron SARS-CoV-2 infections offer low protection against Omicron infection, both for BA.1 and BA.2. Booster vaccination increases protection, although to a lesser extent than against Delta infection and with significant waning within months after the booster. This underlines the importance of timing of future booster doses for high-risk individuals when COVID-19 surges are expected. Further, the high number of infections by Omicron BA.1 and BA.2 escaping infection- or vaccine-induced immunity could contribute to broader population immunity in the months to come.

## Methods

**Laboratory data.** Data from two large diagnostic laboratories were used, which analyse specimens from national community testing in the Netherlands and make use of the TaqPath COVID-19 RT-PCR Kit (ThermoFisher Scientific). This PCR-kit tests for three targets (S, ORF1ab and N). S-gene target failure (SGTF) in combination with a proper signal from ORF1ab and N, also referred to as S-drop-out, has proven to be a highly specific proxy for SARS-CoV-2 variants containing the 69/70 deletion in spike and therefore for Omicron BA.1 when this variant emerged[13]. With lower viral loads, the S-gene target tends to be less sensitive and therefore only positive results with a Ct value of ≤ 30 for the ORF1ab and N targets were included for further analyses. A proper signal in all three targets is a proxy for the Delta or Omicron BA.2 variant, depending on the time period, as these variants do not contain the 69/70 deletion in spike. We defined two cohorts: cohort Delta-Omicron BA.1 with test data from 22 November 2021 to 7 January 2022 and cohort Omicron BA.1-BA.2 with test data from 26 January 2022 to 31 March 2022 (Fig. 1). These time periods were based on a positive predictive value (PPV) of SGTF (with WGS data as reference)[2] of >85% to discern the different variants. The SGTF-PCR testing in the two laboratories covered 20.9% of all tests done at community testing centers in the Netherlands during the study periods and cover the majority of Dutch regions.

**Epidemiologic data.** From 1 June 2020 onwards, mass community testing for SARS-CoV-2 organized by the 25 regional Public Health Services (PHS) has been available and advised for Dutch citizens experiencing COVID-19-like symptoms. During the Delta-Omicron BA.1 cohort period, close contacts of persons testing positive were additionally advised to test as soon as possible and on day 5 after the last contact, irrespective of symptoms. Since 18 January 2022, during the Omicron BA.1-BA.2 cohort period, close contacts with either booster vaccination or recent infection (from 1 January 2022) were exempt from this advice. Test results from the two laboratories performing SGTF-PCR testing were linked to the national community testing register (CoronIT) using a unique sample number. The national community testing register contains pseudonymized data with demographic characteristics, self-reported vaccination status, presence of symptoms at time of requesting the test and symptom onset date if applicable. Regarding vaccination status, we had information on the number of doses, and the date and brand of the last dose received. Specifically for Janssen vaccine, since January 2022 we had information on earlier vaccination with Janssen based on self-report, which enabled us to discern primary vaccination with a 2-dose schedule and booster vaccination after a 1-dose Janssen schedule. We excluded tests if confirmation of a positive self-administered antigen test was the reason for testing. Persons with a positive test 30 days before the current test (the test included in the study) were excluded to avoid including multiple tests of the same episode. Persons with unknown vaccination status were also excluded. Of persons testing more than once during the study period, either the first positive test (if any positive test during the study period) or a randomly selected negative test was included in the analysis to only include one test per person.

**Vaccination and previous infection status definitions.** The previous infection was defined as a positive test by PCR or antigen test at the PHS testing locations at any time point between 1 June 2020 and 30 days before the test included in the study. Primary vaccination was defined as having received two doses of Comirnaty (BNT162b2, BioNTech/Pfizer), Spikevax (mRNA-1273, Moderna) or Vaxzevria (ChAdOx1, AstraZeneca) more than 14 days before the symptom onset date or one dose of Janssen vaccine more than 28 days before the symptom onset date. Booster vaccination was defined as having received a third dose at least 7 days before symptom onset or a second dose after Janssen, if the date of last vaccination was after 18 November 2021 (the start of the booster vaccination campaign in the Netherlands). For cases without a reported onset date, sample date minus two days was used (two days is the median time between onset date and sample date for persons in which onset date is known). Persons who did not receive any vaccine were defined as unvaccinated.

To study the protective effect of infection and vaccination, we made seven categories based on self-reported vaccination status and confirmed previous

infection status and we refer to this as 'vaccination and previous infection status' in the remainder of the manuscript. "Naïve" includes unvaccinated persons without a known previous infection, as registered in the national community testing register. "Previous infection, unvaccinated" includes unvaccinated persons with at least one previous infection. "Primary vaccination" includes persons with a completed primary vaccination series without known previous infection. "First infection, then primary vaccination" includes persons with a previous infection where the number of vaccine doses received at the time of the previous infection was zero. "First start primary vaccination, then infection" includes persons with a previous infection where the number of vaccine doses received at the time of previous infection was at least one. "Booster vaccination" includes persons with a booster vaccination without known previous infection. "Previous infection, booster" includes persons with a booster vaccination and a previous infection; this previous infection could be before or after the booster vaccination. Tests for which the vaccination and previous infection status could not be placed in any of these categories were excluded. Vaccination and previous infection status was further stratified into time since last event, which pertains to the date of last vaccination or the sampling date of the last positive test.

**Statistical analysis.** We compared vaccination and previous infection status between persons testing positive for Delta, Omicron BA.1 or Omicron BA.2 (based on SGTF) and persons testing negative. We performed multinomial logistic regression with vaccination and previous infection status as the independent variable and test result (negative as reference level, positive SGTF, positive non-SGTF) as the dependent variable, adjusting for testing date (as natural cubic spline with 4 degrees of freedom), 5-year age group, sex and PHS region (25 levels). We calculated the relative reduction of odds of a positive test result, with vaccination and previous infection status "naïve" as reference, after vaccination (i.e. vaccine effectiveness) and previous infection as (1-*odds ratio*)*100. Vaccination and previous infection status was further categorized by time since last event (either infection or vaccination) into 30-day categories to study potential waning of the protection of infection or vaccination over time. We also calculated the effectiveness of booster vaccination relative to primary vaccination by using "Primary vaccination" as the reference instead of "Naïve".

Children <18 years of age were excluded from the main analysis comparing all vaccination and previous infection statuses, because they were not yet eligible for (booster) vaccination in the study period. Children <18 years old were included in the analyses stratified by age group (<=11, 12–17, 18–29, 30–59, ≥60 years), to assess the protective effect of the previous infection against infection by variant and, for 12–17 years old, to assess effectiveness of primary vaccination. Vaccinated children aged 5–11 years were excluded from the analysis because they were only eligible for primary vaccination in a part of the study period. Interaction terms between age group and vaccination and previous infection status were added to the model to assess significance for differences between age groups.

Data was analysed using R (version 4.1.2) with the tidyverse (version 1.3.1) R package collection. For statistical analysis the VGAM (version 1.1–5) and splines (version 4.1.2) packages were used.

**Ethical statement.** The Centre for Clinical Expertise at the National Institute for Public Health and the Environment (RIVM) assessed the research proposal following the specific conditions as stated in the law for medical research involving human subjects. The work described was exempted for further approval by the ethical research committee. Pathogen surveillance is a legal task of the RIVM and is carried out under the responsibility of the Dutch Minister of Health, Welfare and Sports. The Public Health Act provides that RIVM may receive pseudonymised data for this task without individual consent.

**Reporting summary.** Further information on research design is available in the Nature Research Reporting Summary linked to this article.

## Data availability

The data underlying Fig. 1 are deposited in Data S4. The relative reduction data underlying Figs. 2 and 3 generated in this study have been deposited in the Data S1 and S2 files. The raw case-based data are protected and are not available due to data privacy laws. On request aggregated data is available but not with the level of detail as used in the analysis because of potential for identifiability of individuals. Contact the corresponding author for access requests (mirjam.knol@rivm.nl) with a response to request timeframe of 3 weeks. The WGS data used in this study are available in the GISAID database under accession IDs found in the GISAID Acknowledgment Table (DOI: doi.org/10.55876/gis8.220701en, GISAID Identifier: EPI_SET_20220701en).

## Code availability

Code for data processing, statistical analysis, figures, and tables can be found at GitHub (github.com/Stijn-A/SARS-CoV-2_Protection_SGTF_Delta_Omicron_BA1_BA2) and Zenodo (https://doi.org/10.5281/zenodo.6670320).

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

## Acknowledgements

The authors would like to thank all personnel at the 25 Public Health Services, and the members of the RIVM COVID-19 surveillance and epidemiology team: Agnetha Hofhuis, Anne Teirlinck, Anne-Wil Valk, Bronke Boudewijns, Carolien Verstraten, Claudia Laarman, Femke Jongenotter, Fleur Petit, Guido Willekens, Irene Veldhuijzen, Jan Polman, Jan van de Kassteele, Janneke van Heereveld, Janneke Heijne, Kirsten Bulsink, Liselotte van Asten, Liz Jenniskens, Lieke Wielders, Loes Soetens, Maarten Mulder, Maarten Schipper, Marit de Lange, Naomi Smorenburg, Nienke Neppelenbroek, Patrick van den Berg, Priscila de Oliveira Bressane Lima, Rolina van Gaalen, Senna van Iersel, Siméon de Bruijn, Sjoerd Wierenga, Susan Lanooij, Sylvia Keijser, Tara Smit, Thomas Dalhuisen, Don Klinkenberg, Jantien Backer, Pieter de Boer, Scott McDonald, Amber Maxwell, Annabel Niessen, Danytza Berry, Daphne van Wees, Henri van Werkhoven, Eric Vos, Frederika Dijkstra, Jeanet Kemmeren, Kylie Ainslie, Albert Jan van Hoek, Birgit van Benthem, Eveline Geubbels, Jacco Wallinga, Rianne van Gageldonk-Lafeber. This work was funded by the Ministry of Health, Welfare and Sports (VWS).

## Author contributions

S.P.A., Bd.G., M.J.K. designed the study; Nv.M., L.A., B.V., R.S. generated the data; S.P.A., Bd.G., D.E., M.J.K. performed analyses; S.J.M.H., C.B.E.M.R., H.E.M., S.vd.H., M.J.K. supervised the study; S.P.A., Bd.G., C.vd.E., M.J.K. wrote the manuscript with input from all authors. All authors reviewed the manuscript.

## Competing interests

The authors declare no competing interests.
