## [Peer Review File · Nature Communications]

Protection of COVID-19 vaccination and previous infection against Omicron BA.1, BA.2 and Delta SARS-CoV-2 infectionsREVIEWER COMMENTS

Reviewer #1 (Remarks to the Author):

Overall.

This is a well written manuscript with important findings. However there are some issues that need to be addressed. The selection procedure for controls might influence the results and I suggest to perform an additional analyses to investigate this, my concerns are described under Methods.

VE against omicron is increasing with age after booster vaccination, previous infection booster and for first start primary vaccination then infection? Several groups 60+ have a very high VE against omicron compared to younger age groups, which is unexpected. These findings should be discussed.

As I understand the description of the vaccine roll-out the oldest were vaccinated first, therefore it is difficult to conclude about the VE and age in figure 2.

Background

Line 53-55: You give the coverage for those 18 years and above, what about those 12- 17 years?

Method:

Line 72 is called Study population, however line 73-83, is a description of the laboratory method not a description of the study population, suggest to rename this section. In this section you write that data come from two large laboratories, what proportion of all tests did they cover? And is the population they cover representative of the national population?

Line 84-95 is a description of the test strategy and data organization.

Line 87-90: you write "When a test is requested, a questionnaire is taken including questions on COVID-19 vaccination status. Test results from the two laboratories were linked to the national community testing register (CoronIT), containing pseudonymized data with demographic characteristics and vaccination status." I am not sure I understand line 87-90, vaccination status is self reported and then when the test result is linked to the community testing register then demographic data and vaccination status is available. Are there two sources for vaccination status and how do you link, a unique identifier or something else?

Line 86-87 you write: Individuals with a positive result on a self-administered rapid antigen test were also urged to seek confirmation PCR testing. But in line 90-91 you write. "We excluded tests performed to confirm a positive self-administered antigen test". This is a strange procedure, the antigen tests can be very unspecific and it will introduce a bias if false positive/negative tests are not corrected. In addition, this sentence in line 91-92 is unclear "and persons with a positive test 30 days before the current test to avoid including multiple tests of the same episode" were excluded. But in a test negative design you only include the first positive test so you do not include multiple tests. This should be clarified.

Line 93-96: "Of persons testing more than once during the study period, either the first positive test (if any positive test during the study period) or the last negative test was included in the analysis to only include one test per person." You write that you have mass testing for SARS-CoV-2, I suppose this means that during the study period several individuals were tested many times?, so why do you chose to take the last negative test? You could have taken the first negative test or a random test among all the negative test a person might have? I think this is a very important issue to look into, because if several controls are tested several times you end up selecting many negative tests taken by the end of the study period, which create a low percentage positive test by the end of the study period where omicron is dominant. I do not think controlling for testing day by a cubic spline can account for this potential bias. I think the way to select controls must be further investigated and if you already did please describe the impact of this selection in the manuscript.

Line 99-100: what about individuals who received a mix of vaccines e.g. Pfizer and Moderna. In Figure S1 they are not listed as excluded?

Line 101-102: where does symptoms come from, are they collected in the questionnaire?

Line 109-121: definition of the immune status "booster" is missing

Line 123-127: there is no dependent variable defined? I suppose it should be test result. You adjust for testing date but you need to check if the way you select your controls impact your VE

estimates. The analysis estimating VE when accounting for time since last event is not described in the method section.

Results

Line 140: the omicron number is not the same as in the table.

Line 140-142: there is a high proportion of omicron during the end of the study where a large proportion of your negative controls most likely also are selected, which must be checked (see previous comments).

Line 141-142. There is an early increase in 18-29 years of age but not in 12-29 years of age as written

Line 143-156: it is a little misleading to directly compare and discuss VE from primary vaccination and booster (Table 2) as time since vaccination is not controlled for in these estimates. I suggest to make a combined description from Table 2 and Figure 1.

Line 154-156: it is unclear where these relative reduction numbers come from

Line 157-164: In the text you refer to the relative reduction in protection presented in Figure 1, I suggest the number behind the figure is shown in a supplementary table or else it is difficult to follow the numbers mentioned in the text.

Line: 161-162 you write "Also in persons with both primary vaccination and previous infection, waning of protection was observed by time since vaccination or infection" This is not the case for delta?

Line: 162-164: ". The protection shortly after booster vaccination was comparable to protection shortly after a combination of primary vaccination and infection (60-70%)." You could add, But higher for both omicron and delta after only previous infection or only primary vaccination. This is how it looks based on Figure 1.

Line 165-166: "Protection conferred by previous infection against Omicron BA.1 was higher for children 0-11 and 12-17 compared to persons aged 18-59, especially for Omicron BA.1." based on Figure 2, this is only the case for omicron?

Line 167-168: "The effectiveness of primary vaccination decreased with increasing age for both variants," I think this is comparing apples and oranges, the oldest were vaccinated first so the waning shown in Figure 1 should be accounted for.

In figure 2, it is strange that VE against omicron is increasing with age after booster vaccination, previous infection booster and for first start primary vaccination then infection? Several groups 60+ have a very high VE against omicron, which is unexpected. For delta, age does not seem to influence booster VE, or VE after a combined infection and vaccination.

Discussion

Line 196-197: you should add: in particular after only previous infection or only primary vaccination

Line 216-217: You write "Highest levels of protection were observed in recently boosted individuals with a previous infection." But this is not different from the level of VE in the groups first infection then vaccination or the other way around, when you compare the same time since last event (Figure 2).

Line 226-228 : , "in our study the previous infections in persons with first infection then vaccination will likely have been infections by wildtype SARS-CoV-2 or the Alpha variant" Was there not a lot of younger people who received their first dose during the summer 2021? They could have had Delta before receiving the vaccines.

Line 226-229: Of note.... What is your conclusion on this "of note"?

Line 230-237: This paragraph is only about severe disease but this study does not present estimates against severe disease. Unless the authors have estimates against severe disease this paragraph could be deleted.

Line 238- 256: Several limitations are mentioned however, the impact of these are not discussed for all of them. It would be good to have an overall discussion of all the limitations

The age specific VE estimates from Figure 2 are not at all mentioned in the discussion, as mentioned in the results there are some unexpected results that the authors needs to address in the discussion. You also need to discuss what impact your choice of controls have.

Reviewer #2 (Remarks to the Author):

The authors assessed vaccine effectiveness of primary vaccination, booster vaccination and the protection conferred by previous SARS-CoV-2 infection against Omicron BA.1 and Delta variants infection, respectively, in the Netherlands from 22 November 2021 to 19 January 2022 when Omicron emerged and replaced Delta. They used a test-negative case-control study design that they applied to a laboratory community-testing database from two large laboratories of the Netherlands that use the TaqPath COVID-19 RT-PCR Kit (ThermoFisher Scientific). Results from the laboratory database were linked to the national community testing register (CoronIT), which contains pseudonymized data with demographic characteristics and vaccination status.

Comments

The objectives of the paper are well justified and the methodology used is in principle sound and (although, there are limitations, see below) to tackle the questions of field effectiveness raised by the authors. The study brings new information to what has already been reported on the effectiveness of primary and booster vaccination, with and without previous infections, by time since vaccination or previous infection and by age group on the risk of infection with Omicron and Delta, respectively. The paper is well written and timely. It certainly contributes to the body of knowledge on this still widely investigated question. The paper is competitive with respect to other studies on this question and brings some new insights (in particular the assessment of the effectiveness of vaccination and previous infection and on the sequence of both for the two variants). The authors discussed appropriately the results in link to the current literature. They acknowledges some of the limitations of the study design. Conclusions are supported by results.

However, several issues need to be addressed by the authors.

Major comments:

1. Authors used laboratory-testing data from two large diagnostic laboratories in the Netherlands. What proportion of Sars-Cov-2 testing do this laboratory account for? What is the representativeness of their testing activities with regards to all testing done in the Netherlands? The testing in these laboratories is for primary care only or not? This information is important for assessing external validity of the results.

2. S-gene target failure (SGTF) in combination with a proper signal from ORF1ab was used to define Omicron infection while a proper signal in all three targets defined Delta variant infection. The authors indicate that this proxy is highly specific. However, for a case-control study design the issue is more on the predictive value positive (PPV) that is dependent of the specificity and the prevalence of the variant in the population. What is the specificity of both proxies for Omicron and Delta? Are they similar? As the proportion of Delta and Omicron influences the PPV for Omicron and Delta, with an increasing prevalence of Omicron and decreasing prevalence for Delta, PPV changed overtime in the opposite direction for both variants. This may result in more misclassification for Delta over time with the opposite for Omicron. How this may have affected the results? This is a limitation to add.

3. The authors indicate that: "Test results from the two laboratories were linked to the national community testing register (CoronIT) containing pseudonymized data with demographic characteristics and vaccination status". How the link between the two databases was done? Using a unique personal identifier (social security number...)? Alternatively, using a set of variables from the two databases? The quality of matching both databases is not similar in one case or the other which. Authors should specify explicitly how they link both databases.

4. Very few co-variables (date of testing and age) were available to take into account potential confounders that have been shown to be important to be taken into account in observatory design (including the TPD design) in particular, place of residence, social inequity status (through an ecological deprivation index, for example)... In addition, matching for place of residence at time of testing takes into account the probability of exposure to the virus (here the authors assess vaccine effectiveness against infection given exposure, which assumes similar exposure to the virus for vaccinated, unvaccinated and previously infected subjects).

5. Cases were community incident infection with either Omicron or Delta with no information on whether they were symptomatic or not. Is that information available in the testing database? At least it is necessary to have information on the proportion of infection with symptoms for both variant during the study period. Since Omicron is less symptomatic than Delta, this may have resulted in some differential selection bias of both variants. The same information is also important for test negative controls: were they always symptomatic?

More minor comments:

6. Line 38: "Several studies suggest Omicron causes less severe disease..." Given data available now, "indicate" would be more appropriate.

7. Line 40: "However this benefit for public health...: The benefit of what? greater transmission or less severity? In addition, can one say this is really a "benefit"? Better to say "characteristic..." In addition, the greater transmission of Omicron leads to more infections and contributes more to population immunity.

8. Line 90: What does "Persons with unknown immune status were also excluded." mean? The reviewer guess authors mean "immunization status", which is clearer to say than "immune status" that refers more to immunological marker testing/monitoring.

9. Line 93: in the same line of thoughts as above, it would be more appropriate to say "Immunization and previous infection status definitions" than "Immune status definitions"

10. Reference 9. The reviewer cannot retrieve the paper through the internet link indicated. Is it the same reference as the paper following paper published in Eurosurveillance:
<https://www.eurosurveillance.org/content/10.2807/1560-7917.ES.2022.27.4.2101196> (

11. The reviewer did not find any reference to ethical approval of this study or to data protection rules. This should be documented.

General changes

As approved by the editor, we added results on the protection of vaccination and previous infection against Omicron BA.2 compared with Omicron BA.1 to the revised manuscript, using the same methodology. Using data from a longer time period, we now have defined two cohorts: cohort Delta-Omicron BA.1 with test data from 22 November 2021 to 7 January 2022 and cohort Omicron BA.1-BA.2 with test data from 26 January 2022 to 31 March 2022. These time periods are now based on a positive predictive value (PPV) of SGTF of >85% to discern the different variants. In the original submission the end of the study period was based on the low Delta incidence. Based on the new definition we shortened the Delta-Omicron BA.1 period as compared to the original submission. The addition of the Omicron BA.1-BA.2 cohort and its results obviously led to large changes in the methods and results section. Additionally, one small change was made. We now consider individuals who received their second dose after the start of the booster campaign with no additional information on whether their first dose was Janssen as having an unknown vaccination status (this could be a booster dose after Janssen or the second primary vaccination dose); in the previous version those individuals were considered primary vaccinated. Conclusions concerning the Delta-Omicron BA.1 comparison in the original submission were unchanged. We also updated the introduction and discussion with (more recent) literature on Omicron BA.1 and BA.2.

Reviewer #1 (Remarks to the Author):

Overall.

This is a well written manuscript with important findings. However there are some issues that need to be addressed. The selection procedure for controls might influence the results and I suggest to perform an additional analyses to investigate this, my concerns are described under Methods.

VE against omicron is increasing with age after booster vaccination, previous infection booster and for first start primary vaccination then infection? Several groups 60+ have a very high VE against omicron compared to younger age groups, which is unexpected. These findings should be discussed.

We agree with the reviewer that some of these results were unexpected. Some of the estimates in the 60+ year age group had quite wide confidence intervals, warranting caution with the interpretation of the results in this age group, but for some comparisons the 60+ group indeed showed higher protection against Omicron BA.1 and BA.2. We now added a paragraph on the differences between age groups in the discussion.

As I understand the description of the vaccine roll-out the oldest were vaccinated first, therefore it is difficult to conclude about the VE and age in figure 2.

We agree with the reviewer that age and time since vaccination are correlated which complicates the interpretation of Figure 2 (in the current version Figure 3). Therefore, we changed Figure 2 (in the current version Figure 3) to contain four different time since event groups (start-59 days, 60-119 days, 120-179, and 180+ days; the sample size did not permit similar groups as in Figure 1) and an overall estimate.

Background

Line 53-55: You give the coverage for those 18 years and above, what about those 12- 17 years?

The coverage in the 12-17 years age group was 70% on April 3 2022. We added this information to the introduction.

Method:

Line 72 is called Study population, however line 73-83, is a description of the laboratory method not a description of the study population, suggest to rename this section. In this section you write that data come from two large laboratories, what proportion of all tests did they cover? And is the population they cover representative of the national population?

We renamed the section.

The two laboratories cover 20.9% of all tests done at community testing centers in the Netherlands. The two labs processed tests from multiple Dutch health regions over the full study period. Comparing the age and sex distribution of tests performed in the two labs with tests performed in other laboratories shows no relevant differences (see table). We do observe differences in regional representation, dependent on the Public Health Service regions the two laboratories operate for. Testing policies do not differ between regions (as they are implemented on a national level). We added the coverage of the two laboratories to the methods section.

	Cohort	Delta-Omicron BA.1	Delta-Omicron BA.1	Omicron BA.1-BA.2	Omicron BA.1-BA.2
	Laboratory	Included in the study n (%)	Other n (%)	Included in the study n (%)	Other n (%)
n		765793	2480500	1040139	4347249
Age	0-9	95877 (12.5)	276209 (11.1)	66449 (6.4)	245783 (5.7)
	10-19	121116 (15.8)	365182 (14.7)	176955 (17.0)	652181 (15.0)
	20-29	122594 (16.0)	443874 (17.9)	193299 (18.6)	838633 (19.3)
	30-39	121648 (15.9)	441984 (17.8)	176486 (17.0)	794881 (18.3)
	40-49	108629 (14.2)	372494 (15.0)	159069 (15.3)	679640 (15.6)
	50-59	90444 (11.8)	290784 (11.7)	136751 (13.1)	586976 (13.5)
	60-69	62649 (8.2)	178841 (7.2)	79476 (7.6)	335364 (7.7)
	70-79	32466 (4.2)	81379 (3.3)	40136 (3.9)	162930 (3.7)
	80-89	9076 (1.2)	24587 (1.0)	10282 (1.0)	44376 (1.0)
	90+	1076 (0.1)	4124 (0.2)	1150 (0.1)	6032 (0.1)
	Unknown	218 (0.0)	1042 (0.0)	86 (0.0)	453 (0.0)
Sex	Male	368741 (48.2)	1134784 (45.7)	492552 (47.4)	1972921 (45.4)
	Female	396571 (51.8)	1343057 (54.1)	547027 (52.6)	2371567 (54.6)
	Unknown	481 (0.1)	2659 (0.1)	560 (0.1)	2761 (0.1)
Province	Groningen	383 (0.1)	97563 (4.0)	434 (0.0)	167157 (3.9)
	Fryslân	252 (0.0)	111821 (4.5)	555 (0.1)	213046 (4.9)

	Drenthe	137 (0.0)	82169 (3.3)	267 (0.0)	165647 (3.8)
	Overijssel	1685 (0.2)	214611 (8.7)	5580 (0.5)	401681 (9.3)
	Flevoland	1466 (0.2)	75625 (3.1)	17041 (1.7)	105733 (2.5)
	Gelderland	236194 (31.2)	159141 (6.5)	307614 (29.9)	435852 (10.1)
	Utrecht	15825 (2.1)	283207 (11.5)	36702 (3.6)	409597 (9.5)
	Noord-Holland	62899 (8.3)	489197 (19.9)	65044 (6.3)	728201 (16.9)
	Zuid-Holland	4157 (0.5)	630191 (25.6)	47925 (4.7)	967755 (22.5)
	Zeeland	307 (0.0)	82995 (3.4)	986 (0.1)	110271 (2.6)
	Noord-Brabant	261945 (34.6)	202148 (8.2)	336655 (32.7)	466706 (10.8)
	Limburg	172854 (22.8)	31689 (1.3)	209598 (20.4)	138501 (3.2)

Line 84-95 is a description of the test strategy and data organization.

We updated the headings of the sections.

Line 87-90: you write “When a test is requested, a questionnaire is taken including questions on COVID-19 vaccination status. Test results from the two laboratories were linked to the national community testing register (CoronIT), containing pseudonymized data with demographic characteristics and vaccination status.” I am not sure I understand line 87-90, vaccination status is self reported and then when the test result is linked to the community testing register then demographic data and vaccination status is available. Are there two sources for vaccination status and how do you link, a unique identifier or something else?

The data on self-reported vaccination status (based on the questionnaire when a test is requested) is stored in the national community testing register (CoronIT). We rephrased this part to clarify this.

Line 86-87 you write: Individuals with a positive result on a self-administered rapid antigen test were also urged to seek confirmation PCR testing. But in line 90-91 you write. “We excluded tests performed to confirm a positive self-administered antigen test”. This is a strange procedure, the antigen tests can be very unspecific and it will introduce a bias if false positive/negative tests are not corrected.

To be clear, the self-administered antigen tests are not included in CoronIT, only tests performed at the community testing center. So if people went to the community testing center with the reason to confirm their positive self-test, they were excluded from the analysis. Note that only people testing positive on a self-administered test are asked to confirm it, and not people testing negative. We think it would potentially introduce bias when we would include these confirmation tests from persons with a positive self-test result, considering the test-negative study design – the necessary negative tests would be missing from the data. We rephrased this part to clarify this.

In addition, this sentence in line 91-92 is unclear “and persons with a positive test 30 days before the current test to avoid including multiple tests of the same episode” were excluded. But in a test negative

design you only include the first positive test so you do not include multiple tests. This should be clarified.

We indeed include the first positive test of a person in the study period. However, we also looked at previous positive tests (before the study period) to define immune status (now vaccination and previous infection status). In that case, we did not include tests from persons that had a positive test less than 30 days before the 'current' test (the test falling within the study period) in order to prevent testing for the same infection episode. We rephrased this part to clarify this.

Line 93-96: "Of persons testing more than once during the study period, either the first positive test (if any positive test during the study period) or the last negative test was included in the analysis to only include one test per person." You write that you have mass testing for SARS-CoV-2, I suppose this means that during the study period several individuals were tested many times?, so why do you chose to take the last negative test? You could have taken the first negative test or a random test among all the negative test a person might have? I think this is a very important issue to look into, because if several controls are tested several times you end up selecting many negative tests taken by the end of the study period, which create a low percentage positive test by the end of the study period where omicron is dominant. I do not think controlling for testing day by a cubic spline can account for this potential bias. I think the way to select controls must be further investigated and if you already did please describe the impact of this selection in the manuscript.

We find it indeed more logical to use a random negative test instead of the last negative test, so we implemented this in the revised manuscript. This did not change our results. In our study population, around 15% had more than one test during the study period (Supplementary Figure 1).

Line 99-100: what about individuals who received a mix of vaccines e.g. Pfizer and Moderna. In Figure S1 they are not listed as excluded?

We only had information about the last vaccine received, as described in the methods, so we could not discern or exclude mixed schedules. In the Netherlands, generally the same vaccine was used for the primary vaccination series; only Vaxzevria followed by Comirnaty (0.3% of administered primary vaccine schedules) as heterologous primary series was possible.

Line 101-102: where does symptoms come from, are they collected in the questionnaire?

Yes, when requesting a test, persons are asked about their symptoms and onset date. Information on symptoms is only available at the moment of requesting the test. We added this to the methods section. We also present the percentage of persons with and without symptoms in Table 1 and performed a sensitivity analysis including only persons with symptoms (Table S2).

Line 109-121: definition of the immune status "booster" is missing

"Booster vaccination" includes persons with a booster vaccination without known previous infection. We added this to the methods section.

Line 123-127: there is no dependent variable defined? I suppose it should be test result. You adjust for testing date but you need to check if the way you select your controls impact your VE estimates. The analysis estimating VE when accounting for time since last event is not described in the method section.

The dependent variable is indeed the test result. We corrected this in the statistical analysis section. And we added the description of the analysis of VE by time since last event to this section.

Results

Line 140: the omicron number is not the same as in the table.

The correct number was in the table, we corrected the text.

Line 140-142: there is a high proportion of omicron during the end of the study where a large proportion of your negative controls most likely also are selected, which must be checked (see previous comments).

See answer above.

Line 141-142. There is an early increase in 18-29 years of age but not in 12-29 years of age as written

We corrected this.

Line 143-156: it is a little misleading to directly compare and discuss VE from primary vaccination and booster (Table 2) as time since vaccination is not controlled for in these estimates. I suggest to make a combined description from Table 2 and Figure 1.

Indeed, our study period was during a time where most people had received their primary vaccination some time ago, so the estimate of primary vaccination VE is applicable to the situation at that moment. We now added the overall estimate to Figure 2 (former Figure 1) and deleted Table 2. In addition, we added the median time since primary and booster vaccination to Table 1.

Line 154-156: it is unclear where these relative reduction numbers come from

This is the estimate for protection from booster vaccination relative to primary vaccination. We now added this analysis to the methods section.

Line 157-164: In the text you refer to the relative reduction in protection presented in Figure 1, I suggest the number behind the figure is shown in a supplementary table or else it is difficult to follow the numbers mentioned in the text.

We added a supplementary table (Table S2) with the numbers from Figure 1 (in the current version Figure 2) and a supplementary table (Table S4) with the numbers from Figure 2 (in the current version Figure 3).

Line: 161-162 you write “Also in persons with both primary vaccination and previous infection, waning of protection was observed by time since vaccination or infection” This is not the case for delta?

It is indeed less apparent for Delta than for Omicron BA.1. Although with higher levels of protection, it is also more difficult to observe waning. We rephrased this part of the results.

Line: 162-164: “. The protection shortly after booster vaccination was comparable to protection shortly after a combination of primary vaccination and infection (60-70%).” You could add, But higher for both omicron and delta after only previous infection or only primary vaccination. This is how it looks based on Figure 1.

We added this.

Line 165-166:” Protection conferred by previous infection against Omicron BA.1 was higher for children 0-11 and 12-17 compared to persons aged 18-59, especially for Omicron BA.1.” based on Figure 2, this is only the case for omicron?

We do observe significant interaction between previous infection and age 0-11 and 12-17 for BA.1 and BA.2, and just significant for Delta for 0-11 age group (Table S4). For Omicron this is added to the results.

Line 167-168: “The effectiveness of primary vaccination decreased with increasing age for both variants,” I think this is comparing apples and oranges, the oldest were vaccinated first so the waning shown in Figure 1 should be accounted for.

We now present the age-stratified analysis by four different time since event groups (start-59 days, 60-119 days, 120-179, and 180+ days; Figure 3).

In figure 2, it is strange that VE against omicron is increasing with age after booster vaccination, previous infection booster and for first start primary vaccination then infection? Several groups 60+ have a very high VE against omicron, which is unexpected. For delta, age does not seem to influence booster VE, or VE after a combined infection and vaccination.

It is indeed surprising that VE is higher for 60+ in some of the groups. Some of the estimates in the 60+ year age group had quite wide confidence intervals, warranting caution with the interpretation of the results in this age group, but for some comparisons the 60+ group indeed showed higher protection against Omicron BA.1 and BA.2. We now added a paragraph on the differences between age groups in the discussion.

Discussion

Line 196-197: you should add: in particular after only previous infection or only primary vaccination

We added this.

Line 216-217: You write “Highest levels of protection were observed in recently boosted individuals with a previous infection.” But this is not different from the level of VE in the groups first infection then vaccination or the other way around, when you compare the same time since last event (Figure 2).

That is correct, we rephrased this.

Line 226-228 : , “in our study the previous infections in persons with first infection then vaccination will likely have been infections by wildtype SARS-CoV-2 or the Alpha variant” Was there not a lot of younger people who received their first dose during the summer 2021? They could have had Delta before receiving the vaccines.

This is a good point. We rephrased the sentence.

Line 226-229: Of note.... What is your conclusion on this “of note”?

It means that not only the sequence of infection and vaccination differed but most likely also the variant that was encountered previously. So it means that the groups may not be directly comparable and therefore the results are more difficult to interpret. We added this to the discussion.

Line 230-237: This paragraph is only about severe disease but this study does not present estimates against severe disease. Unless the authors have estimates against severe disease this paragraph could be deleted.

Our manuscript is indeed about infections and not severe disease, but we think it is important to also mention the protection against severe disease by the different variants, as this is also very relevant for health care capacity.

Line 238- 256: Several limitations are mentioned however, the impact of these are not discussed for all of them. It would be good to have an overall discussion of all the limitations

We included an overall discussion of the limitations of our study design and we tried to discuss the impact of all potential limitations, although sometimes it is difficult to predict in which direction the bias would operate.

The age specific VE estimates from Figure 2 are not at all mentioned in the discussion, as mentioned in the results there are some unexpected results that the authors needs to address in the discussion. You also need to discuss what impact your choice of controls have.

We now added a paragraph on the results by age group. And we changed the method to select the controls.

Reviewer #2 (Remarks to the Author):

The authors assessed vaccine effectiveness of primary vaccination, booster vaccination and the protection conferred by previous SARS-CoV-2 infection against Omicron BA.1 and Delta variants infection, respectively, in the Netherlands from 22 November 2021 to 19 January 2022 when Omicron emerged and replaced Delta. They used a test-negative case-control study design that they applied to a laboratory community-testing database from two large laboratories of the Netherlands that use the TaqPath COVID-19 RT-PCR Kit (ThermoFisher Scientific). Results from the laboratory database were linked to the national community testing register (CoronIT), which contains pseudonymized data with demographic characteristics and vaccination status.

Comments

The objectives of the paper are well justified and the methodology used is in principle sound and (although, there are limitations, see below) to tackle the questions of field effectiveness raised by the authors. The study brings new information to what has already been reported on the effectiveness of primary and booster vaccination, with and without previous infections, by time since vaccination or previous infection and by age group on the risk of infection with Omicron and Delta, respectively. The paper is well written and timely. It certainly contributes to the body of knowledge on this still widely investigated question. The paper is competitive with respect to other studies on this question and brings some new insights (in particular the assessment of the effectiveness of vaccination and previous infection and on the sequence of both for the two variants). The authors discussed appropriately the results in link to the current literature. They acknowledges some of the limitations of the study design. Conclusions are supported by results.

However, several issues need to be addressed by the authors.

Major comments:

1. Authors used laboratory-testing data from two large diagnostic laboratories in the Netherlands. What proportion of Sars-Cov-2 testing do this laboratory account for? What is the representativeness of their testing activities with regards to all testing done in the Netherlands? The testing in these laboratories is for primary care only or not? This information is important for assessing external validity of the results.

The two laboratories cover 20.9% of all tests done at community testing centers in the Netherlands. The two labs processed tests from multiple Dutch health regions during certain time periods, so this should be quite random. Comparing the age and sex distribution of tests performed in the two labs with tests performed in other laboratories shows no relevant differences (see table). We do observe differences in regional coverage, dependent on the Public Health Service regions the two laboratories operate in. Testing policies do not differ between regions (as they are implemented on a national level). We added the coverage of the two laboratories to the methods section.

The tests included in this study are all from community testing centers, so it does not include tests performed in hospitals.

	Cohort	Delta-Omicron BA.1	Delta-Omicron BA.1	Omicron BA.1-BA.2	Omicron BA.1-BA.2
	Laboratory	Included in the study n (%)	Other n (%)	Included in the study n (%)	Other n (%)
n		765793	2480500	1040139	4347249
Age	0-9	95877 (12.5)	276209 (11.1)	66449 (6.4)	245783 (5.7)
	10-19	121116 (15.8)	365182 (14.7)	176955 (17.0)	652181 (15.0)
	20-29	122594 (16.0)	443874 (17.9)	193299 (18.6)	838633 (19.3)
	30-39	121648 (15.9)	441984 (17.8)	176486 (17.0)	794881 (18.3)
	40-49	108629 (14.2)	372494 (15.0)	159069 (15.3)	679640 (15.6)
	50-59	90444 (11.8)	290784 (11.7)	136751 (13.1)	586976 (13.5)
	60-69	62649 (8.2)	178841 (7.2)	79476 (7.6)	335364 (7.7)
	70-79	32466 (4.2)	81379 (3.3)	40136 (3.9)	162930 (3.7)
	80-89	9076 (1.2)	24587 (1.0)	10282 (1.0)	44376 (1.0)
	90+	1076 (0.1)	4124 (0.2)	1150 (0.1)	6032 (0.1)
	Unknown	218 (0.0)	1042 (0.0)	86 (0.0)	453 (0.0)
Sex	Male	368741 (48.2)	1134784 (45.7)	492552 (47.4)	1972921 (45.4)
	Female	396571 (51.8)	1343057 (54.1)	547027 (52.6)	2371567 (54.6)
	Unknown	481 (0.1)	2659 (0.1)	560 (0.1)	2761 (0.1)
Province	Groningen	383 (0.1)	97563 (4.0)	434 (0.0)	167157 (3.9)
	Fryslân	252 (0.0)	111821 (4.5)	555 (0.1)	213046 (4.9)
	Drenthe	137 (0.0)	82169 (3.3)	267 (0.0)	165647 (3.8)
	Overijssel	1685 (0.2)	214611 (8.7)	5580 (0.5)	401681 (9.3)
	Flevoland	1466 (0.2)	75625 (3.1)	17041 (1.7)	105733 (2.5)
	Gelderland	236194 (31.2)	159141 (6.5)	307614 (29.9)	435852 (10.1)
	Utrecht	15825 (2.1)	283207 (11.5)	36702 (3.6)	409597 (9.5)
	Noord-Holland	62899 (8.3)	489197 (19.9)	65044 (6.3)	728201 (16.9)
	Zuid-Holland	4157 (0.5)	630191 (25.6)	47925 (4.7)	967755 (22.5)
	Zeeland	307 (0.0)	82995 (3.4)	986 (0.1)	110271 (2.6)
	Noord-Brabant	261945 (34.6)	202148 (8.2)	336655 (32.7)	466706 (10.8)
	Limburg	172854 (22.8)	31689 (1.3)	209598 (20.4)	138501 (3.2)

2.S-gene target failure (SGTF) in combination with a proper signal from ORF1ab was used to define Omicron infection while a proper signal in all three targets defined Delta variant infection. The authors indicate that this proxy is highly specific. However, for a case-control study design the issue is more on the predictive value positive (PPV) that is dependent of the specificity and the prevalence of the variant in the population. What is the specificity of both proxies for Omicron and Delta? Are they similar? As the proportion of Delta and Omicron influences the PPV for Omicron and Delta, with an increasing prevalence of Omicron and decreasing prevalence for Delta, PPV changed overtime in the opposite direction for both variants. This may result in more misclassification for Delta over time with the opposite for Omicron. How this may have affected the results? This is a limitation to add.

Using variant typing from whole genome sequencing we calculated the PPV and sensitivity/specificity for both cohorts and we included this information in the text and as a supplementary table (Table S1). Indeed with changing incidence of specific variants, PPV changes over time. We have now used a cut-off of 85% for the PPV to discern variants to define the cohorts in the new analysis. As a result, the first cohort (study period in original submission) was shortened.

Cohort	SGTF result	WGS result Omicron BA.1	WGS result Delta	WGS result Omicron BA.2	PPV	Sensitivity
Cohort Delta-Omicron BA.1	Not detected	132	0	0	1.00	1.00
Cohort Delta-Omicron BA.1	Detected	0	353	0	1.00	1.00
Cohort Omicron BA.1-BA.2	Not detected	158	0	0	1.00	0.99
Cohort Omicron BA.1-BA.2	Detected	2	0	128	0.98	1.00

3.The authors indicate that: “Test results from the two laboratories were linked to the national community testing register (CoronIT) containing pseudonymized data with demographic characteristics and vaccination status”. How the link between the two databases was done? Using a unique personal identifier (social security number...)? Alternatively, using a set of variables from the two databases? The quality of matching both databases is not similar in one case or the other which. Authors should specify explicitly how they link both databases.

The databases were linked using a unique sample number. We added this to the methods section.

4.Very few co-variables (date of testing and age) were available to take into account potential confounders that have been shown to be important to be taken into account in observatory design (including the TPD design) in particular, place of residence, social inequity status (through an ecological deprivation index, for example)... In addition, matching for place of residence at time of testing takes into account the probability of exposure to the virus (here the authors assess vaccine effectiveness against infection given exposure, which assumes similar exposure to the virus for vaccinated, unvaccinated and previously infected subjects).

We now added Public Health Service region (the Netherlands is divided in 25 Public Health Service regions) of residence to the model, which did not impact the results. Unfortunately, the available level of detail on place of residence is not detailed enough to assign an ecological deprivation index, so we could not adjust for any measure of socio-economic status.

5. Cases were community incident infection with either Omicron or Delta with no information on whether they were symptomatic or not. Is that information available in the testing database? At least it is necessary to have information on the proportion of infection with symptoms for both variant during the study period. Since Omicron is less symptomatic than Delta, this may have resulted in some differential selection bias of both variants. The same information is also important for test negative controls: were they always symptomatic?

When requesting a test, persons are asked about their symptoms and onset date. Information on symptoms is only available at the moment of requesting the test. We added this to the methods section and now also present the percentage of persons with and without symptoms at the time of test request in Table 1. In addition, we performed a sensitivity analysis including only persons with symptoms. The results did not substantially change. We added the results to a supplementary table (Table S2).

More minor comments:

6. Line 38: "Several studies suggest Omicron causes less severe disease..." Given data available now, "indicate" would be more appropriate.

We changed this.

7. Line 40: "However this benefit for public health...: The benefit of what? greater transmission or less severity? In addition, can one say this is really a "benefit"? Better to say "characteristic..." In addition, the greater transmission of Omicron leads to more infections and contributes more to population immunity.

We mean the benefit of less severity here. We changed this to 'characteristic'.

8. Line 90: What does "Persons with unknown immune status were also excluded." mean? The reviewer guess authors mean "immunization status", which is clearer to say than "immune status" that refers more to immunological marker testing/monitoring.

We mean that we excluded persons with unknown vaccination status. We changed this. Furthermore, we now use vaccination and previous infection status instead of immune status to refer both to immunity from infection and immunity from vaccination.

9. Line 93: in the same line of thoughts as above, it would be more appropriate to say "Immunization and previous infection status definitions" than "Immune status definitions"

We chose to use vaccination and previous infection status to denote both immunity from infection and immunity from vaccination and we explained this the first time the term was used.

10. Reference 9. The reviewer cannot retrieve the paper through the internet link indicated. Is it the same reference as the paper following paper published in Eurosurveillance:
<https://www.eurosurveillance.org/content/10.2807/1560-7917.ES.2022.27.4.2101196> (

Yes, this is the correct reference. We corrected this.

11. The reviewer did not find any reference to ethical approval of this study or to data protection rules. This should be documented.

We added an ethical statement to the manuscript.

REVIEWERS' COMMENTS

Reviewer #1 (Remarks to the Author):

The authors have considered and addressed all my comments and concerns and I agree that including a longer time period has improved the manuscript. I do not have any further comments.

Reviewer #2 (Remarks to the Author):

The authors have responded appropriately to reviewers comments and ammended and improved their manuscript. The fact they now distinguish 2 time cohort period to track BA.1 vs BA.2 variants and assess VE accordingly is an important and timely improvement. The reviewer noticed that adjustment for confounding remained limited by lack of more precise data available.